# System for Detecting Learner Stuck in Programming Learning

**DOI:** 10.3390/s23125739

**Published:** 2023-06-20

**Authors:** Hiroki Oka, Ayumi Ohnishi, Tsutomu Terada, Masahiko Tsukamoto

**Affiliations:** Graduate School of Engineering, Kobe University, 1-1 Rokkodaicho, Nada, Kobe 657-8501, Hyogo, Japan; hiroki-oka@stu.kobe-u.ac.jp (H.O.); ohnishi@eedept.kobe-u.ac.jp (A.O.); tuka@kobe-u.ac.jp (M.T.)

**Keywords:** programming learning, stuck, multi-modal, sensing, heart rate information, machine learning

## Abstract

Getting stuck is an inevitable part of learning programming. Long-term stuck decreases the learner’s motivation and learning efficiency. The current approach to supporting learning in lectures involves teachers finding students who are getting stuck, reviewing their source code, and solving the problems. However, it is difficult for teachers to grasp every learner’s stuck situation and to distinguish stuck or deep thinking only by their source code. Teachers should advise learners only when there is no progress and they are psychologically stuck. This paper proposes a method for detecting when learners get stuck during programming by using multi-modal data, considering both their source code and psychological state measured by a heart rate sensor. The evaluation results of the proposed method show that it can detect more stuck situations than the method that uses only a single indicator. Furthermore, we implemented a system that aggregates the stuck situation detected by the proposed method and presents them to a teacher. In evaluations during the actual programming lecture, participants rated the notification timing of application as suitable and commented that the application was useful. The questionnaire survey showed that the application can detect situations where learners cannot find solutions to exercise problems or express them in programming.

## 1. Introduction

As the use of information technology advances, understanding programs has become increasingly important. Several countries are offering classes on programming and emphasizing concepts such as “Computational Thinking [1]” in their education. In elementary education, block-based programming environments such as Scratch [2], micro:bit [3], and LEGO Mindstorms [4] are often used. In addition, the number of learning materials related to programming is increasing [5].

Although many programming learning materials and services are available, various barriers to learning programming [6,7,8,9] cause beginners to get stuck in programming. Such a stuck situation, where the learner does not know what to do and needs help, should be supported by others such as a teacher.

Generally, such a stuck situation is resolved by teachers or by computer-based tools to assist the learner, such as Intelligent Tutoring Systems (ITSs) or Adaptive Learning Systems (ALSs). ITSs are learning support systems that autonomously provide feedback to learners, and it has been reported that ITSs provide hints for problem-solving in learning, enabling efficient learning [10,11]. ALSs are learning support systems for personalizing learning based on learners’ progress and interests and has the potential to improve learning outcomes in computer science education [12]. However, it is difficult for teachers and such learning systems to identify when learners are stuck and need assistance. For example, when the learner’s typing stops, he/she does not know how to solve the problem, and if he/she is in a stalemate, we should help him/her overcome getting stuck. On the other hand, even if the type has stopped, if the learner is concentrating on solving the problem, the intervention of the teacher or the learning system may prevent learning. Therefore, it is necessary to consider what kind of psychological state the learner is in to resolve getting stuck that should be supported.

Previous studies have estimated programmers’ perceived task difficulty, emotions, and progress based on data collected from programmers [13,14,15]. However, no study has estimated when a learner that is becoming stuck should be assisted in learning programming. In addition, no programmer state estimation system has been proposed that considers both the source code and the psychological state of the learner. Furthermore, no real-time stuck detection system has been implemented.

In this study, we focus on heart rate information, which is an indicator of psychological states such as stress and concentration, and propose a method to detect getting stuck by using both the learner’s heart rate information and source code. In addition, to verify whether the proposed method is useful in actual programming lectures, we implemented a detection application for getting stuck using the proposed method and conducted evaluation experiments in programming lectures for beginners.

The detection method of getting stuck presented in this paper is the same as that proposed in our previous conference paper [16]. The result data of the experiment in Section 3 are also the same. The novelty of this paper compared to our previous conference paper is that we have implemented a real-time stuck notification web application and conducted an evaluation experiment. The contributions of this paper are as follows.

This paper proposes a method for detecting when learners become stuck during programming by using multi-modal data, considering both their source code and psychological state measured by a heart rate sensor.We implemented a real-time web application that notifies learners when they are stuck and evaluated it in a programming class. The evaluation results showed that the timing of the notifications was appropriate and that participants evaluated the application as useful.

## 2. Related Works

In this section, we introduce research on the difficulty of learning programming that is a source of getting stuck in programming and describe research on estimating programmer state by logging data such as the source code and biometric data.

### 2.1. Difficulties in Learning Programming

There are various difficulties in learning programming that can cause a learner to become stuck. From the perspective of cognitive load theory, Sweller stated that the more information one must process simultaneously, the higher the cognitive load becomes, making learning more difficult [17]. Learning programming is no exception and various difficulties in learning programming cause individuals to become stuck. These difficulties are not only superficial ones caused by the program itself, such as errors, but many of them are also related to internal conditions, such as the learner’s level of understanding and thinking. In particular, many studies have been conducted on the difficulties faced by beginners in learning programming. Derus et al. conducted a questionnaire survey of students taking a basic programming course [6]. They investigated the difficulties in learning programming and found that the most difficult topics were understanding abstract concepts such as multidimensional arrays, iterations, and functions, as well as understanding the structure of programs and designing programs to solve certain problems. Milne et al. surveyed students and teachers in an introductory object-oriented programming course about the concepts and topics that students find difficult [7]. They found that students had difficulty understanding concepts related to pointers and memory and that this was due to their inability to construct mental models of memory behavior in their programs. Lahtinen et al. conducted a questionnaire survey of over 500 students and teachers about the difficulties in learning programming [8]. The results showed that the most difficult tasks were “understanding how to design a program to solve a certain problem”, “dividing a function into steps”, and “finding bugs in one’s programs”. The most difficult concepts were abstract concepts such as “recursion”, “pointers and references”, and “abstract data types”. Ismail et al. interviewed five computer engineering teachers about problems in programming education and found that the four main causes of problems students face in learning programming are (1) lack of problem analysis skills, (2) ineffective problem-solving representations, (3) ineffective teaching methods for problem solving and programming, and (4) inability to understand and master program syntax and structure [9].

Therefore, learners’ being stuck is caused by a variety of difficulties, many of which are related to the learner’s state, making it difficult to determine whether or not to help the learner resolve the stuck when he/she becomes stuck.

### 2.2. Research on Source Code Analysis

Several studies have been conducted to predict program quality and potential bugs by analyzing programmers’ source code. Nagappan et al. showed that several program complexity indices could be used to predict post-release software defects in different software projects [18]. They also found that no single metric can predict defects in all software projects. Such complexity metrics are also used for learning programming. Truong et al. proposed a static analysis framework that provides feedback to beginning Java students on how to write better programs based on metrics used in software engineerings such as cyclic complexity and structural analysis of programs using AST (Abstract Syntax Tree) [19].

Based on these works, it is possible to predict latent defects in the source code based on indicators related to the program such as cyclic complexity and AST of the source code. However, these indicators do not work with only a single indicator, and they are not applicable to source code containing syntax errors.

### 2.3. Research on State Estimation during Programming Using Biometric Data

Several studies estimate programmer status by logging biometric data during programming. Crosby et al. analyzed programmers’ gaze when reading programs and found that skilled programmers focused on important parts of the program and complex instructions, while programming beginner tended to focus on comments and comparisons [20]. Uwano et al. implemented a system called DRESREM to measure the eye movements of reviewers during source code review [21]. Using this system, they analyzed eye movements during source code review and found that there is a pattern called “scan”, in which the reviewer looks at the entire code from top to bottom and then focuses on details. The result showed that reviewers who did not spend enough time scanning took longer to find defects in the source code. Siegmund et al. used fMRI to analyze the brain during program comprehension and found that brain regions related to working memory and language processing were activated [22].

Some studies use multiple biometric data. Lee et al. used EEG and eye movements to estimate programmer expertise and program comprehension task difficulty [13]. The results showed that programmer expertise and task difficulty were estimated with 97.7% and 64.9% accuracy, respectively. Fritz et al. measured programmers’ eye movements, skin electrical activity, and electroencephalography to estimate the difficulty level of a program comprehension task [23]. The results showed that the difficulty level was estimated with an accuracy of 70.0%. Müller et al. estimated the developer’s emotion and progress based on skin electrical activity, electroencephalography, skin temperature, heart rate, blood volume pulse rate, and pupil diameter [14]. They were able to classify developers’ emotions as positive or negative with 71.3% accuracy and their progress as high or low with 67.7% accuracy.

Biometric data such as eye gaze, brain waves, and heartbeats can be used to estimate the programmer’s understanding and emotions. However, in an actual programming learning environment, biometric measurement devices should not interfere with learning. Therefore, in this study, we detect programmers being stuck using a heart rate sensor, which has a low wearing load and does not interfere with programming.

### 2.4. Research on Programmer State Estimation by Combining Multiple Indicators

Several studies have attempted to estimate the programmer’s state by combining several measures obtained during programming. Ishida et al. used the programmer’s EEG and the referenced document type to estimate the programmer’s comprehension level [24]. In their study, students majoring in computer science were given a task to read a Java program and were able to classify task success or failure with 85.2% accuracy based on their EEG and the document type they referenced. Carter et al. classified programmer activities into five categories (Navigation, Focus, Edit, Debug, and Remove) and used these categories in combination with the programmer’s posture captured by the Microsoft Kinect to detect situations in which the programmer subjectively feels difficulty [15].

Combining multiple indicators obtained during programming can provide a more accurate picture of the programmer’s situation. However, it is difficult to detect when a learner is stuck in learning programming. Programmers are not always aware that they are stuck, and even if they are experiencing difficulties, if they can solve them on their own to deepen their understanding, others should not intervene. There has been no research aimed at detecting getting stuck situations in which learners need assistance in real-time in the actual learning environment. In this study, we detect situations in which beginning programmers become stuck and need assistance by using both the learner’s heart rate information and the source code.

## 3. Proposed Method

In this section, we propose a method for detecting when an individual is stuck during programming using heart rate information and source code and describe its details.

### 3.1. Assumed Environment

In this study, we assume an environment in which the learner is working on a programming task. In this environment, we obtain heart rate information from a heart rate sensor and source code from a text editor. Using these data, we create detection features and detect when the learner is stuck.

In this paper, we define the state of getting stuck as “a state in which the programming learner does not know what to do and needs help”. Such a stuck state depends on the psychological state of the learner, such as whether he/she is concentrating or not and whether he/she is thinking or not. In a programming lecture, the output of the learner’s thinking is expressed in the source code, and by reading it, teachers can grasp the learning status based on information such as whether or not the subject is satisfied and whether or not it can be executed without errors. However, it is difficult to identify if a student is stuck only from the source code written by the learner, and if others give unnecessary advice or hints, they may interfere with the learner’s thinking and concentration. Therefore, our purpose is to detect situations in which the learner is stuck and needs help by using the learner’s source code together with heart rate information, which is an indicator of psychological state.

### 3.2. Stuck Detection Method

The proposed detection method detects the state of learners being stuck by inputting the heart rate information of the learner and metrics calculated from the source code (code-related metrics) as features to the classifier. The proposed method and an example of its use case are shown in the Figure 1. We use Random Forest [25] as a classifier. The specific features used are described below.

#### 3.2.1. Heart Rate Information

We use LF/HF and pNN50 as indicators of psychological states such as concentration and stress in learners. The LF/HF value indicates the balance between the sympathetic and parasympathetic nervous systems, and the higher the value, the more dominant the sympathetic activity is, resulting in high concentration and high stress. The pNN50 is the percentage of heartbeats in which the difference between adjacent heartbeat peaks exceeds 50 ms and is an index of vagal tone intensity. The lower the value, the higher the concentration and stress. Because a certain amount of data length is required to calculate LF/HF, we use pNN50 as an immediate measure and LF/HF as a long-term measure.

#### 3.2.2. Code-Related Metrics

McCabe’s complexity [26] and Halsted’s complexity [27] are used in software development as indices to evaluate the maintainability and potential bugs of large programs. However, this study assumes that the size of the program is relatively small and that it is an assignment for a programming lecture. Therefore, we use the source lines of code (SLOC), the AST edit distance from the previous program execution, and the elapsed time from the previous program execution as indicators. AST is a structure that extracts only the semantically important parts of a program and expresses them in a tree structure. For example, in JavaScript, the AST specification is defined as an ESTree [28]. We used each indicator as a measure of the amount of code written by the learner, the change in the structure of the code, and the frequency of execution.

### 3.3. Evaluation Experiment of the Stuck Detection Method

We conducted an evaluation experiment to assess whether the proposed method can detect thet state of being stuck in learning programming. In the experiment, we assumed a situation in which learners solved a task in an introductory programming lecture using visual outputs and had the participants solve the task of reproducing a given image by executing a program. We used JavaScript as the programming language and p5.js [29] as the drawing library. We also used Esprima as a parser for JavaScript programs [30]. The images used for each task used in the experiment are shown in Figure 2a,b. The image used in Task 1 was designed to use iteration and conditional branching, and the image used in Task 2 was designed to use iteration, functions, and recursion. The source codes of example solutions for each task are shown in Codes 1 and 2.

The participants were five men in their 20 s (Participants A–E). The experiment procedure is shown in Figure 3. First, after a 10 min explanation of the basics of JavaScript necessary for solving the tasks, the participants worked on the first task for 40 min. After a 10 min break, the participants worked on the second task for 40 min. After the experiment, we conducted a questionnaire survey.
Code 1: Example code of Task11const s = 50;2const d = 40;3const n = 8;4
5function setup() {6  createCanvas(400, 400);7  background(255);8  noStroke();9
10  for (let i = 0; i < n; i++) {11    for (let j = 0; j < n; j++) {12      const cx = i * s;13      const cy = j * s;14      if ((i + j) % 2 === 0) {15        fill(122);16        rect(cx, cy, s);17        if (j < 3) {18          fill(255, 0, 0);19        } else if (4 < j) {20          fill(0);21        }22        ellipse(cx + s/2, cy + s/2, d);23      }24    }25  }26}
Code 2: Example code of Task 2.
1const w = 2;2let cells3let generation = 0;4const ruleset [0, 0, 0, 1, 0, 1, 1, 0];5  6function setup() {7  createCanvas(512, 256);8  background(255);9  cells = new Array(floor(width/w)).fill(0);10  cells[cells.length/2] = 1;11  noStroke();12  fill(0);13}14  15function draw(){16  for (let i = 0; i < cells.length; i++){17    if (cells[i] === 0) rect(i * w, generation * w, w, w);18  }19if (generation < height/w) generate();20}21  22function generate(){23  const nextgen = new Array(cells.length).fill(0);24  for (let i = 1; i < cells.length-1; i++) {25    nextgen[i] = rules(cells[i-1], cells[i], cells[i+1]);26  }27  cells = nextgen;28  generation++;29}30
31function rules(a, b, c) {32  if (a == 1 && b == 1 && c == 1) return ruleset[0];33  if (a == 1 && b == 1 && c == 0) return ruleset[1];34  if (a == 1 && b == 0 && c == 1) return ruleset[2];35  if (a == 1 && b == 0 && c == 0) return ruleset[3];36  if (a == 0 && b == 1 && c == 1) return ruleset[4];37  if (a == 0 && b == 1 && c == 0) return ruleset[5];38  if (a == 0 && b == 0 && c == 1) return ruleset[6];39  if (a == 0 && b == 0 && c == 0) return ruleset[7];40  return 0;41}


During the experiment, we attached a WHS-3 heart rate sensor [31] to the participants and collected heart rate information. We also collected code-related metrics from the participants’ text editors. Based on the collected data, we created a model using heart rate information, code-related metrics, and both of these as features (hereafter referred to as the Biometric model, Code model, and Multi-modal model, respectively) and compared their accuracy in classifying stuck. The first author used a labeling tool, ELAN [32], to create correct labels. While observing the participant’s face and screen images during the experiment, the author labeled the part where the participant was stuck using the source code, changes in the output, and the participant’s expressive behavior as keys. Specifically, we considered actions such as expressions (e.g., tilting the head or touching the face) to be stuck-related actions, and then subjectively judged whether the source code or the program execution result was close to the correct answer and labeled them as stuck/not stuck.

The experiments described in this section were conducted with the approval of the Ethical Review Committee for Research Directly Involving Human Subjects of the Graduate School of Engineering, Kobe University (approval numbers 04-13).

### 3.4. Evaluation Result of the Stuck Detection Method

We compared the detection accuracy of each model by inter-trial cross-validation, with each task being one trial. All data were collected every second. Task 1 had 2985 data labeled stuck and 6335 data labeled not stuck, and Task 2 had 3837 data labeled stuck and 6476 data labeled not stuck. Therefore, when training the model, we used SMOTE [33], which is an oversampling method and corrected the data bias by increasing the number of data that were labeled stuck.

To compare the correspondence between the stuck classified by the three models and the correct stuck, we introduced a post-processing algorithm that decides that if 2/3 of the data is label as stuck in the last 60 s, the learner is stuck at that point. The results are shown in Figure 4 and Figure 5. The total number of stuck parts for all participants in the experiment was 20. The word “stuck part” means a chunk of data labeled “stuck”. The Biometric model showed only 10 stuck parts corresponding to actual stuck parts when applying the post-processing algorithm due to the rapidly changing classification results. The Code model showed 15 stuck parts corresponding to actual stuck parts, and the Multi-modal model showed 17 stuck parts corresponding to actual stuck parts.

The detection accuracy of Code and Biometric models was 62%, and that of Multi-modal model was 63%. Table 1, Table 2 and Table 3 show the confusion matrix for the classification results of each model. The labels in the top row represent data classified by the model as stuck/not stuck, and the labels in the left column represent ground truth data labeled as stuck/not stuck during labeling. The table shows that the Multi-modal model can detect the most number of data labeled as stuck and has the highest F-score (0.37).

From these results, using heart rate information and code-related metrics together can detect more stuck than using only a single indicator.

## 4. Stuck Notification Application

This section describes the implementation and evaluation experiments of a stuck notification application using the proposed method described in Section 3.

### 4.1. System Design and Implementation

To use the proposed method in a programming seminar course, we implemented an application that notifies the learners of being stuck to the teachers. The application detects the learners being stuck based on the learner’s heart rate and code-related metrics and visualizes it to the teachers. Because the application is implemented as a web application, multiple teachers can use it on their respective terminals by accessing the URL from a web browser. The application screen and data flow are shown in Figure 6 and Figure 7.

First, the application retrieves the source code and stores it in the database whenever the user saves it. We use Visual Studio Code and its extension to retrieve the source code. Heart rate information obtained from the WHS-1 [34], a heart rate sensor worn by the user, is wirelessly transmitted to a back-end PC and written to a CSV format every 40 milliseconds. Then, input features are calculated from these data and the stuck situation is classified using the proposed method, followed by post-processing to store the judgment of each user being stuck in a database. In the post-processing, if 2/3 of the classification results in the past 60 s are classified as stuck, that point is judged as stuck. The status of each user can be checked from a web browser.

The application screen displays each user’s ID and the results of detection by the Code model and the Multi-modal model in a tabular form. In the row for each user ID, the cell corresponding to the column labeled “Code” or “Multi” represents the status, with a red cell indicating being stuck and a white cell indicating being not stuck.

### 4.2. Evaluation

We conducted an evaluation experiment to determine whether the notification application that implements the proposed method is useful in a real programming learning environment. In this experiment, we used the application in a programming seminar course with learners and teachers to evaluate the suitability of notifications. We asked participants to answer a questionnaire in an actual lecture. The teachers conducted the seminar course while checking the notification application and were instructed to talk to the learner when the learner was notified of being stuck. When the notification was made, the teacher evaluated the suitability of the timing of notifying stuck, and the learners evaluated the suitability of the teacher’s timing of talking on a 5-point scale. In addition, participants were asked to select from the following seven options to answer what type of situation they were in.

I cannot think of a way to solve the problem (S1);I cannot program what I want to do (S2);I do not know why, but the program does not work well (S3);The program works, but I do not understand why it works (S4);I do not know what to do (S5);There are no problems or unknowns (S6);Other (S7).

The duration of the experiment was 90 min, during which the participants worked on four to five programming tasks. The tasks in each environment are listed below:

#### 4.2.1. Experiment Environment 1


**Fourth lecture**


Q1: Find the average, maximum, and minimum values of an array of random positive integers and plot them as a bar graph.Q2: Draw a line graph of the values of an array of random positive integers.Q3: Plot the values of an array of random positive integers in a pie chart. In each graph, the value is displayed when the cursor is hovered over it.Q4:-Draw multiple spheres of random size.-Make the number of spheres increase at regular intervals.-Make the spheres bounce back when they collide.-Recreate snowfall.


**Fifth lecture**


Q1: Draw the EU flag.Q2: Draw a regular n-gon.Q3: Draw a string in a speech bubble.Q4: Create a function to find the sum of an array.Q5:-Create a function that returns the number of days in any given year.-Create a function that returns what day of the week is m-month d, year y, A.D.-Create a program to draw a calendar for the year y and month m.

#### 4.2.2. Experiment Environment 2

Q1: Reproduce Figure 8a programmatically.Q2: Reproduce Figure 2a programmatically (same as in Section 3.2).Q3: Reproduce Figure 8b programmatically.Q4: Reproduce any flag programmatically.

**Figure 8 sensors-23-05739-f008:**
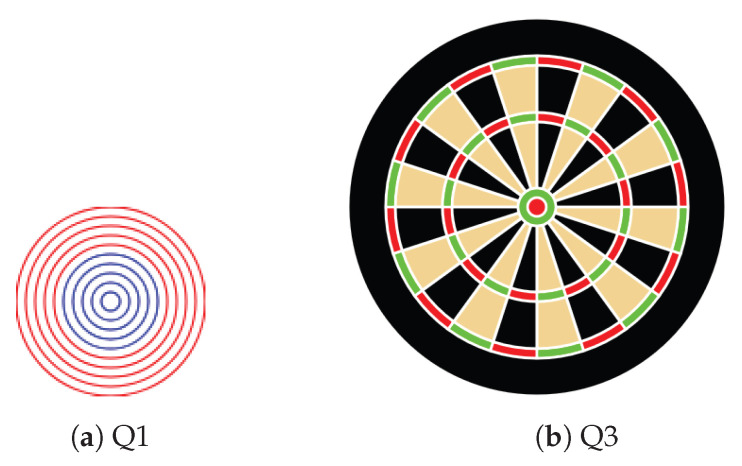
Images used in Experiment Environment 2.

The programming language used was JavaScript. After the experiment, the participants were asked to complete a questionnaire about the application.

The experiment was conducted in two different environments: (1) in an actual lecture and (2) in a laboratory. Each environment is shown in Figure 9a,b. Environment 1 was the “Basic Programming Exercise 1” lecture at the Faculty of Global Human Sciences, Kobe University. The participants were six students and three teachers, and the experiment was conducted in the fourth and fifth lectures. In Experiment Environment 2, the participants were nine male students in their twenties from the Faculty of Engineering and one teaching assistant.

The experiments described in this section were conducted with the approval of the Ethical Review Committee for Research Directly Involving Human Subjects of the Graduate School of Engineering, Kobe University (approval numbers 04-13 and 04-21).

### 4.3. Evaluation Result

We describe the results obtained in two experiment settings. Because data were missing for one participant in each of the two experiment environments, the results are presented for the 13 participants (Participant A-M) for whom data were available. In some cases, participants were incorrectly judged to have gotten stuck when they moved on to the next task, so the data and evaluations for these cases are omitted.

Table 4 and Table 5 show the evaluation of the notification timing obtained for each experiment environment. These tables show that the participants rated the notification timing of application as suitable and indicate that the application is useful in a programming lecture. In addition, we found that the suitability ratings did not differ much between the learner and the teacher. However, there was a difference in the number of notifications handled by the Multi-modal model and the Code model, and the two models could not be compared.

Furthermore, the table indicates that the learner was often in S1 or S2 at the time of notification. Such a stuck situation occurs not only in programming tasks with visual output but also in tasks that implement algorithms for solving certain problems or implementing software. Therefore, we believe that the application can be applied to other programming learning content as well.

Figure 10 shows the results of the stuck notifications in Experiment Environment 2 and the timing at which the teacher responded to the notification. In this figure, the evaluation of the teachers and the learners are averaged, with blue dots indicating a value of less than three and red dots indicating a value of three or more. This figure shows that there are fewer notifications for the Multi-modal model than for the Code model. The number of notifications was generally large, and the teachers were not able to respond to all of them.

Next, we describe the results of a questionnaire survey conducted after the experiment. Table 6 and Table 7 show the results of the learner questionnaire. The results show that LQ1 was rated high and LQ2 was rated low in both experiment environments, indicating that the teachers’ talking to the learners did not interfere with their learning. The ratings of LQ3 and LQ4 indicate that the notification application worked to some extent. The following comments were obtained from the learners indicating that the system is useful in actual lectures. “I thought it was a good system that the TA came to support me even if I didn’t tell them I was having trouble”, “The TA came to me when I didn’t know what to do. I thought it was a good system for students who have difficulty asking questions or who do not know what to do”. “When I was not sure whether to ask a question or not, the TAs came to help me even if I did not call them, so I think it was easier than in a regular class”.

The results of the teachers’ questionnaire after the experiment are shown in Table 8 and Table 9. Regarding Experiment Environment 1, TQ1 and TQ4 received positive evaluations, but TQ2 and TQ3 did not. This was because the teachers had to respond to questions from learners other than the participants, and the load on the teacher was increased by checking the application notifications and responding to the participants who got stuck. Teachers commented that “It took me a long time to respond to questions, and when I got a notification from the app, I was in the middle of responding to a question, or I was taking so many questions from participants that I couldn’t get to the others”, “I thought it was quite useful when the number of learners was small and they could do their tasks on their own to a certain extent, but when there are a lot of learners it can be difficult”.

## 5. Discussion

In this section, based on the results obtained in the evaluation experiment, we describe the effects and improvements of the proposed method and the implementation application, as well as future prospects.

### 5.1. Summary of Experiments in Section 3 and Section 4

In the proposed method, we used LF/HF values and pNN50 obtained from heart rate information, SLOC, AST edit distance, and elapsed seconds obtained from the source code as features of the classifier to detect being stuck. In addition, we implemented a real-time web application that notifies learners when they are stuck. Experiment results of the application showed that the application could be effective in helping teachers monitor learners’ progress. The experiment result also showed that the suitability of the notification was highly rated, and the questionnaire survey showed that the teacher’s talking to the learners did not interfere with their learning. Therefore, the detection accuracy of the proposed method is sufficient to support the teacher in notifying them of learners getting stuck in a small group programming lecture with visual output, such as in this experiment environment.

### 5.2. Comparison of the Result with Related Research

Compared to related studies that use binary classification of states such as programmer difficulty, progress, and emotion [13,14,15], our proposed method is not directly comparable in accuracy because it targets different states. However, unlike other studies, the target of detection in this paper is stuck that the learner needs help with. Merely detecting the difficulty level is not enough to help the learner resolve the stuck situation. In this study, we implemented an application that notifies the teacher of the learner getting stuck, and in an evaluation experiment, we were able to notify the teacher of the learner getting stuck at appropriate times during the lecture and provide support in resolving the situation.

### 5.3. Limitation

The evaluation experiments of the proposed method and the implemented application were conducted with university students. However, if the proposed method and application are used for programming learners at the level of elementary, junior high, and high school students, their points of being stuck may be completely different from those in the present study, such as not understanding multiplication or not being able to read English words used in programming languages. Therefore, the proposed stuck detection method and application could not be applicable to learners at other learning levels. In addition, the evaluation experiment was conducted with a relatively small number of learners (five to nine). In the case of a larger lecture, the number of learners and teachers will vary, and it is necessary to verify the effectiveness of the application in such a situation.

In the evaluation of the proposed method, the first author labeled by looking at the participants’ screen and face video. For more reliable labeling, it is necessary to merge the labels created by multiple people to create the correct label.

### 5.4. Future Work

According to the results of the application evaluation, the number of notifications was large and the teachers were not able to respond to all of them. Therefore, we should eliminate the notifications that erroneously detected changes in tasks as being stuck, and only notify serious situations of being stuck to the extent that teachers can respond to them. In addition, it can be sufficient to display notifications only for the Muti-modal model for use in actual lectures. We also need to improve the user interface, for example, by displaying information on how long the learners have been stuck.

In this study, we implemented the proposed method as an application for notifying the learner of being stuck, assuming a programming lecture with visual output. We believe that this method can be applied in various ways. Because there is no teacher in a self-study environment, it is not possible to ask for guidance from a teacher to resolve getting stuck. Therefore, it is necessary to provide support such as learning the content of being stuck and presenting information according to the content of being stuck to resolve them. Moreover, although we limited the stuck detection to programming tasks with visual output, we believe that the proposed stuck detection method can be applied to other tasks as well. In the future, we will verify whether the proposed method can be applied to programming with text-based output, such as algorithms with standard input/output, and to programming exercises with hardware, such as microcomputers, that deal with complex content.

## 6. Conclusions

In this study, we proposed a method for stuck detecting when learners get stuck during programming by using multi-modal data, considering both their source code and psychological state measured by a heart rate sensor. The evaluation results of the proposed method show that it can detect more stuck than the one using only a single indicator. In addition, we implemented a real-time web application that aggregates the stuck detected by the proposed method and presents them to a teacher. The evaluation results showed that the timing of the notifications was appropriate and that participants evaluated the application as useful.

## Figures and Tables

**Figure 1 sensors-23-05739-f001:**
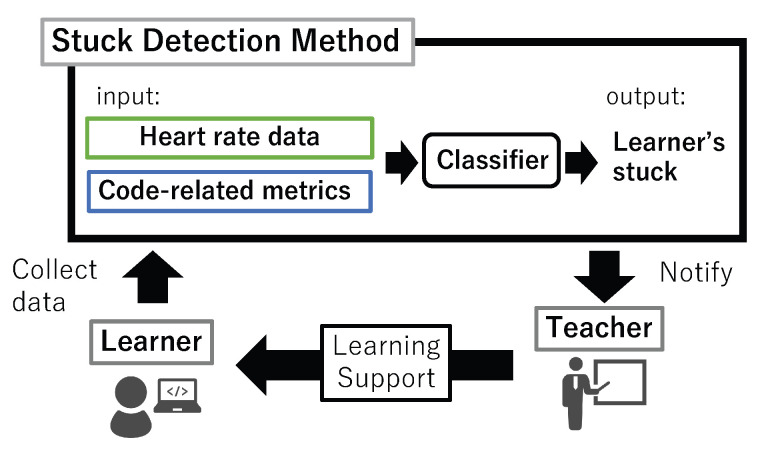
Proposed method and use case in lectures.

**Figure 2 sensors-23-05739-f002:**
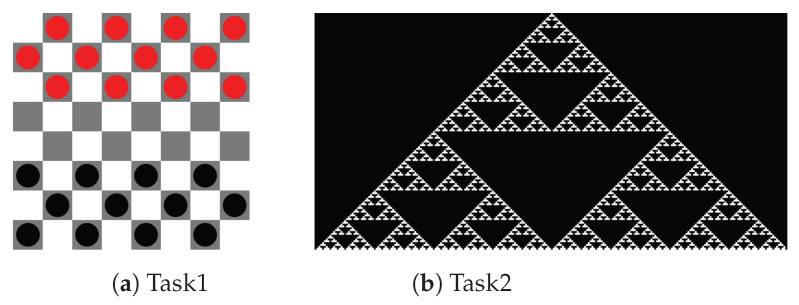
Images used for each task.

**Figure 3 sensors-23-05739-f003:**
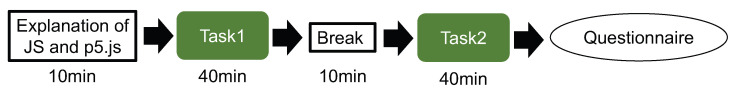
Procedure.

**Figure 4 sensors-23-05739-f004:**
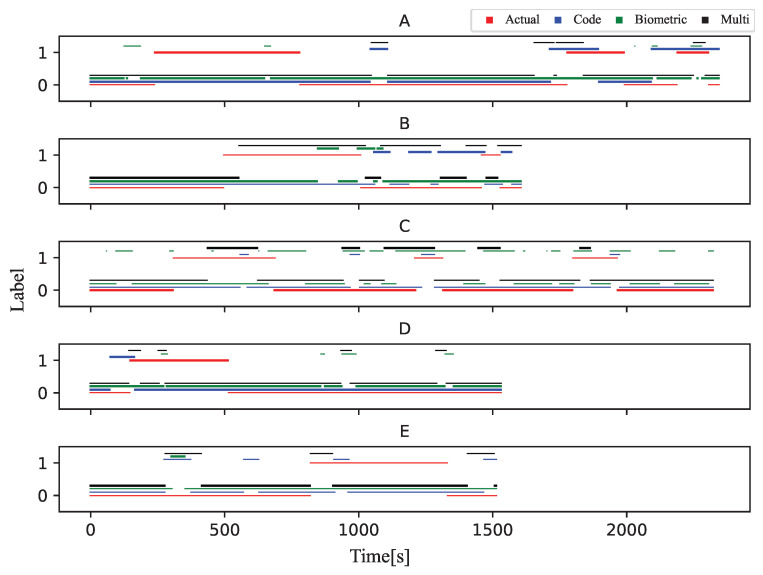
Stuck detection results of Task 1 (Four types of data are displayed stacked vertically).

**Figure 5 sensors-23-05739-f005:**
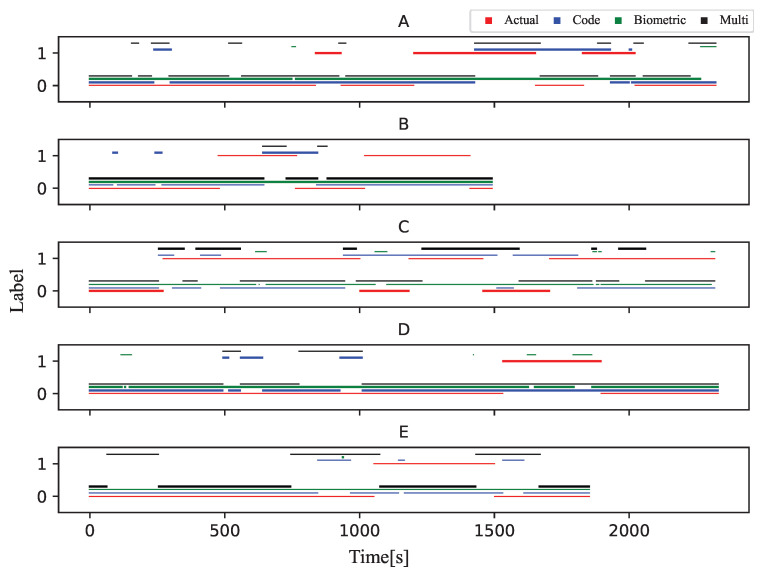
Stuck detection results of Task 2 (Four types of data are displayed stacked vertically).

**Figure 6 sensors-23-05739-f006:**
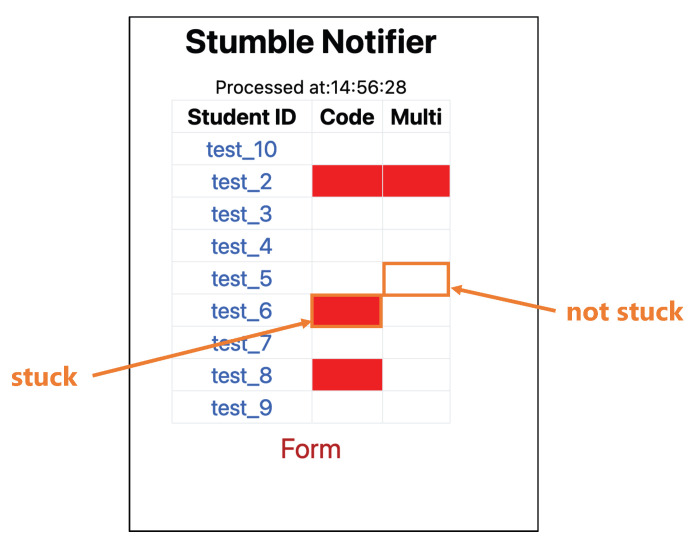
Screen of the application.

**Figure 7 sensors-23-05739-f007:**
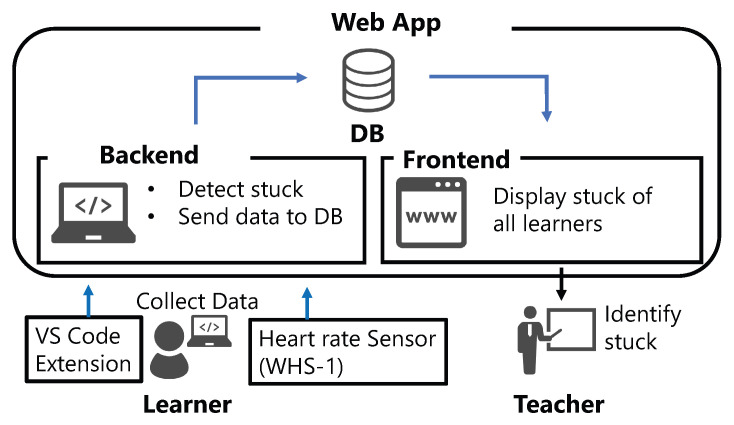
Data flow of the application.

**Figure 9 sensors-23-05739-f009:**
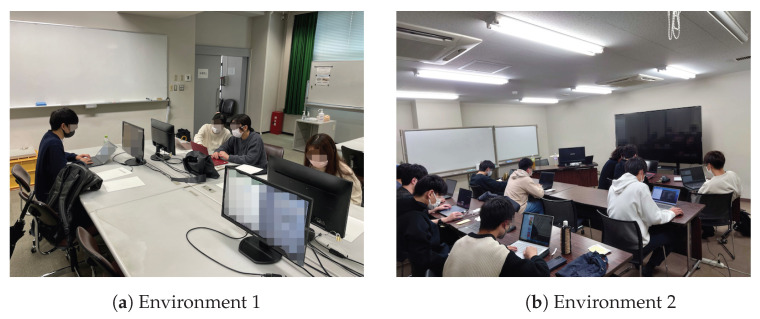
Images of experiment environments.

**Figure 10 sensors-23-05739-f010:**
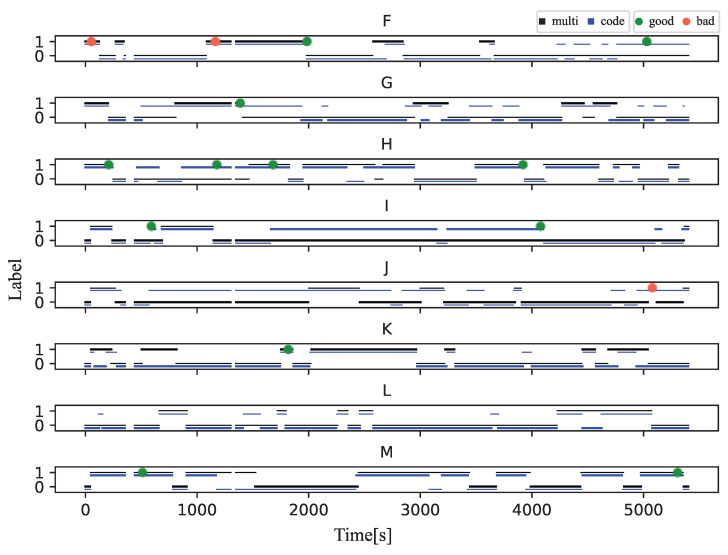
Environment 2: Stuck detection results and response timing.

**Table 1 sensors-23-05739-t001:** Confusion Matrix (Multi-modal).

		**Prediction**		
		not stuck	stuck	**Recall**
**Actual**	not stuck	10,222	4706	0.68
	stuck	2579	2116	0.45
**Precision**		0.80	0.31	0.37

**Table 2 sensors-23-05739-t002:** Confusion Matrix (Code).

		**Prediction**		
		not stuck	stuck	**Recall**
**Actual**	not stuck	10,755	5370	0.67
	stuck	2046	1452	0.42
**Precision**		0.84	0.21	0.28

**Table 3 sensors-23-05739-t003:** Confusion Matrix (Biometric).

		**Prediction**		
		not stuck	stuck	**Recall**
**Actual**	not stuck	11,626	6258	0.65
	stuck	1175	564	0.32
**Precision**		0.91	0.08	0.13

**Table 4 sensors-23-05739-t004:** Environment 1: evaluation of notification timing (1: not suitable–5: suitable).

Participant	Model	Suitability		Situation
		Learner	Teacher	
B	Multi	5	5	S1
A	Code	4	4	S1
A	Code	-	2	-
C	Code, Multi	4	3	S1, S2
D	Code, Multi	3	1	S2
D	Code, Multi	5	5	S1

**Table 5 sensors-23-05739-t005:** Environment 2: evaluation of notification timing (1: not suitable–5: suitable).

Participant	Model	Suitability		Situation
		Learner	Teacher	
F	Code, Multi	3	1	S2
H	Code, Multi	5	4	S1
M	Code, Multi	4	5	S3
I	Code	5	3	S1
H	Code	5	5	S1
F	Code, Multi	3	2	S6
G	Code, Multi	4	4	S2
H	Code, Multi	5	5	S1
K	Code, Multi	5	4	S2
F	Code, Multi	5	5	S2
H	Code, Multi	5	5	S1
I	Code	4	5	S2
F	Code	5	5	S2, S3
J	Code, Multi	-	1	-
M	Code, Multi	5	5	S2

**Table 6 sensors-23-05739-t006:** Environment 1: Questionnaire result of the learners (1: disagree, 5: agree).

Code	Question	AVE	SD
LQ1	Were you able to concentrate in lecture?	4.67	0.52
LQ2	Did the teacher’s talking to you interfere with your concentration?	1.83	0.98
LQ3	Did a teacher come to you when you were in need?	4.00	0.63
LQ4	Would you like to use this system in the future?	3.67	0.82

**Table 7 sensors-23-05739-t007:** Environment 2: Questionnaire result of the learners (1: disagree, 5: agree).

Code	Question	AVE	SD
LQ1	Were you able to concentrate in lecture?	4.67	1.05
LQ2	Did the teacher’s talking to you interfere with your concentration?	1.78	1.09
LQ3	Did a TA come to you when you were in need?	3.22	1.09
LQ4	Would you like to use this system in the future?	3.33	1.12

**Table 8 sensors-23-05739-t008:** Environment 1: Questionnaire result of the teachers (1: disagree, 5: agree).

Code	Question	AVE	SD
TQ1	Was the timing of notification appropriate?	3.50	0.58
TQ2	Have you gained a better grasp of the learners’ learning situation?	3.25	0.96
TQ3	Did the lecture go more smoothly?	2.75	0.50
TQ4	Would you like to use this system in the future?	4.00	0.00

**Table 9 sensors-23-05739-t009:** Environment 2: Questionnaire result of the teachers (1: disagree, 5: agree).

Code	Question	Evaluation
TQ1	Was the timing of notification suitable?	3
TQ2	Have you gained a better grasp of the learners’ learning situation?	5
TQ3	Did the lecture go more smoothly?	4
TQ4	Would you like to use this system in the future?	3

## Data Availability

Ethical restriction.

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
