# Peer review of "System for Detecting Learner Stuck in Programming Learning"

_sensors, 2023, doi:10.3390/s23125739_

Round 1

Reviewer 1 Report

The topic of this paper is quite interesting. The model and tool proposed are highly beneficial in computer science education, particularly in assisting learners when they encounter difficulties during programming.

However, the results regarding the study were not clearly presented. For example, regarding the statement in Line 243, "The total number of sections labeled as a stuck section is 20 for all participants," I am unsure about the exact meaning of "20." Does it refer to a specific unit of measurement or time? Additionally, what is meant by a "section" in this context?

The statement in Line 244 "The Biometric model could only handle 10 stuck sections" is somewhat unclear. It's not clear what constitutes a "stuck section." Could you provide more clarification?

I find Table 1, 2, and 3 to be ambiguous. For instance, I'm unsure about "10,222." It appears in both the "Actual-not stuck" and "Prediction-not stuck" columns simultaneously. Could you please explain how readers should interpret this?

Furthermore, in line 320, it appears that Figure 9a should be referred to as Figure 8a instead.

Finally, regarding Table 4 and 5, since there are only 4 and 9 participants, it may not be appropriate to generalize the suitability score to other situations. The same concern applies to Table 6-9 concerning the questionnaire results. Therefore, I am skeptical that these data alone can support the authors' conclusions regarding the effectiveness of the proposed method in notifying teachers about learners getting stuck during lectures. Particularlly, "The experiment result also showed that the suitability of the notification was highly rated, and the questionnaire survey showed that the teacher’s talking to the learners did not interfere with their learning. Therefore, the detection accuracy of the proposed method is sufficient for supporting teachers in notifying them of learners getting stuck in lectures." -- More evidence is needed for drawing such conclusion. 

Reviewer 2 Report

The paper proposes a method for detecting when learners get stuck during programming by using multi-modal data. The proposed method can detect more stuck than the one using only a single indicator. The authors implemented a system that aggregates the stuck detected by the proposed method and presents them to a teacher. The results of the questionnaire survey showed that the application can detect situations where learners cannot find solutions to exercise problems or express them in programming.

The paper is very well organized. The Literature review section provides sufficient background and includes relevant references. The research design is appropriate. The proposed method and Stuck Notification Application are well described. 

I recommend the authors to compare their results with such of other studies in the field.

Minor editing reuqired

Reviewer 3 Report

This research is quite interesting to me and provides implications for helping students learn better in computer programming. I'd like to recommend a couple of comments to the authors.

First, you mentioned about ITS in the introduction, but nowadays ALS(Adaptive Learning System) is more heeded in educators using AI. They are quite similar but not the same. In my opinion, I guess the system you introduced here is more like ITS, not ALS. You have to refer to some articles related to ALS if you want to discuss more broadly.

Second, the awareness of learners' difficulties is one of crucial issues in learning science and cognitive science and you can cite more influential articles dealing with learners' difficulties.

While reading this article, I could not understand the intention of several sentences, just check out if all sentences are appropriate.
